# Identification of Critical Genes and Signaling Pathways in Human Monocytes Following High-Intensity Exercise

**DOI:** 10.3390/healthcare9060618

**Published:** 2021-05-22

**Authors:** Pengda Li, Li Luo

**Affiliations:** School of Physical Education and Sports Science, Soochow University, Suzhou 215021, China; 20194206027@stu.suda.edu.cn

**Keywords:** high-intensity exercise, monocytes, differentially expressed genes, biological pathways, chronic inflammation

## Abstract

Background: Monocytes are critical components, not only for innate immunity, but also for the activation of the adaptive immune system. Many studies in animals and humans have demonstrated that monocytes may be closely associated with chronic inflammatory diseases and be proved to be pivotal in the association between high-intensity exercise and anti-inflammation response. However, the underlying molecular mechanisms driving this are barely understood. The present study aimed to screen for potential hub genes and candidate signaling pathways associated with the effects of high-intensity exercise on human monocytes through bioinformatics analysis. Materials and Methods: The GSE51835 gene expression dataset was downloaded from the Gene Expression Omnibus database. The dataset consists of 12 monocyte samples from two groups of pre-exercise and post-exercise individuals. Identifying differentially expressed genes (DEGs) with R software, and functional annotation and pathway analyses were then performed with related web databases. Subsequently, a protein–protein interaction (PPI) network which discovers key functional protein and a transcription factors-DEGs network which predicts upstream regulators were constructed. Results: A total of 146 differentially expressed genes were identified, including 95 upregulated and 51 downregulated genes. Gene Ontology analysis indicated that in the biological process functional group, these DEGs were mainly involved in cellular response to hydrogen peroxide, response to unfolded protein, negative regulation of cell proliferation, cellular response to laminar fluid shear stress, and positive regulation of protein metabolic process. The top five enrichment pathways in a Kyoto Encyclopedia of Genes and Genomes (KEGG) pathway analysis were the FoxO signaling pathway, protein processing in the endoplasmic reticulum, influenza A, the ErbB signaling pathway, and the MAPK signaling pathway. *TNF*, *DUSP1*, *ATF3*, *CXCR4*, *NR4A1*, *BHLHE40*, *CDKN1B*, *SOCS3*, *TNFAIP3*, and *MCL1* were the top 10 potential hub genes. The most important modules obtained in the PPI network were performed KEGG pathway analysis, which showed that these genes were mainly involved in the MAPK signaling pathway, the IL-17 signaling pathway, the TNF signaling pathway, osteoclast differentiation, and apoptosis. A transcription factor (TF) target network illustrated that *FOXJ2* was a critical regulatory factor. Conclusions: This study identified the essential genes and pathways associated with exercise and monocytes. Among these, four essential genes (*TNF*, *DUSP1*, *CXCR4*, and *NR4A1*) and the FoxO signaling pathway play vital roles in the immune function of monocytes. High-intensity exercise may improve the resistance of chronic inflammatory diseases by regulating the expression of these genes.

## 1. Introduction

Monocytes are circulating blood leukocytes that are involved in the innate immune response to inflammation [1]. Monocytes, which originate from progenitors in the bone marrow, are part of the first line of immune defense along with neutrophils and circulate in the vasculature, bone marrow, and spleen. In the normal course of development, monocytes migrate into peripheral tissues and differentiate into dendritic cells or macrophages depending on the local cytokine environment [2]. During pathogen challenge, monocytes are mobilized from the bone marrow and recruited to sites of inflammation, where they carry out their respective functions in promoting inflammation or anti-inflammatory responses depending on stimulus style [1]. As part of the mononuclear phagocyte system, monocytes link the innate and adaptive immune responses and mediate antimicrobial host defense and the removal of apoptotic cell debris [3]. Furthermore, monocytes play crucial roles in tissue repair and remodeling [4].

Physical exercise provokes an adaptive response and beneficial effects on health through modulation of the immune system [5]. Many studies in animals and humans have demonstrated that exercise profoundly affects the immune system [6]. Exercise acts as a stressor that can elicit different immune responses. The duration and intensity of exercise are widely considered to be critical elements that may positively or negatively influence physical health and immune responses. There is general agreement that repeated moderate-intensity physical activity is beneficial for immune function, reinforcing the anti-oxidative capacity, reducing oxidative stress, and increasing the efficiency of energy generation, and therefore reducing the incidence of inflammatory diseases. In contrast, long-term intensive exercise training can suppress immune function and increase the risk of upper respiratory infection [7]. However, recent evidence suggests that acute bouts of physical exercise can also regulate the immune response. High-intensity exercise can induce the mass production of the anti-inflammation cytokines interleukin-6 and interleukin-10, resulting in a heightened state of immunocompetence [8,9]. This contributes to prophylactic and therapeutic treatment of diseases associated with chronic inflammation [10]. Additionally, high-intensity exercise also causes the number of circulating monocytes to increase dramatically [11]. Thus, monocytes may be closely associated with chronic inflammatory disease and vascular function [12] and have an essential role in the association between physical activity and anti-inflammation responses. Recent studies have shown that exercise results in altered related gene expression in monocytes [13]. However, the possible molecular mechanism for how high-intensity exercise affects monocytic cells remain unclear. Therefore, additional studies are needed to deeply understand the underlying mechanisms and to gather insight into the beneficial effects of exercise on physical function affecting monocytes.

During the research, the GSE51835 gene expression profile was obtained from the Gene Expression Omnibus (GEO) database [12]. Then a gene profile analysis was used to compare before and after exercise samples to recognize differentially expressed genes (DEGs). Gene Ontology (GO) and Kyoto Encyclopedia of Genes and Genomes (KEGG) pathway enrichment analyses were additionally performed. Subsequently, we instituted transcription factor (TF) target network and protein–protein interaction (PPI) network to find crucial genes and signaling pathways and then used these to deduce an underlying molecular mechanism models of exercise revealing the effects of high-intensity exercise on monocytes.

## 2. Materials and Methods

### 2.1. Microarray Data

In this study, the GSE51835 gene expression profile was obtained from the GEO database (https://www.ncbi.nlm.nih.gov/geo/, accessed on 17 Augest 2020.). GSE51835 was uploaded by Radom-Aizik S et al. [12]. The platform of dataset was the GPL570 (HG-U133_Plus_2) Affymetrix Human Genome U133 Plus 2.0 Array. This dataset includes twelve healthy men (mean age 26 ± 0.6 years and body mass index 26 ± 0.8 kg/m^2^, characteristics of the subjects are shown in Appendix A [12]). The participants who actively joined in the elite or competitive level of sports, or were suffering from any chronic illness or medication, were excluded from the study. Each subject performed ramp-type progressive cycle-ergometer exercise to measure maximum oxygen uptake (VO2max) with an expired gas analyzer. At least 48 h but no more than 10 days later, each subject performed ten 2 min bouts of constant work rate cycle-ergometry exercise, with a 1 min rest at interval. Gas exchange was measured breath-by-breath and oxygen uptake was calculated. The work rate was individualized for each subject and was equivalent to 82% of the participants’ VO2max on average. This protocol was designed to simulate naturally occurring patterns of brief high-intensity exercise. The blood samples were collected before and immediately after exercise. Monocytes were isolated from blood, including classical (CD14++/CD16−), nonclassical (CD16++/CD14+) and intermediate (CD16+/CD14++) monocytes. Subjects were asked to refrain from intense physical activity for 48 h before the exercise challenge. The clinical information of the samples in this dataset is shown in Appendix A.

### 2.2. Microarray Data Pre-Processing

The raw gene expression and platform data as Series Matrix files were downloaded for all analyses. The raw data were pre-processed by using R, a free software for statistics and graphics. Data were normalized with the limma package, including format conversion, complementing missing values, assessing the background correction, and log2 transformed using the limma package in R. Subsequently, annotation packs are used to convert probe identifiers to genetic symbols in that platform.

### 2.3. Screening Differentially Expressed Genes and Hierarchical Clustering Analysis

The classic Bayesian method in the limma R package (version 3.11, https://bioconductor.org/packages/release/bioc/html/limma.html, accessed on 25 Augest 2020.) [14] was applied to screen for DEGs from each dataset. Genes with an adjusted *p* < 0.05 and a |log fold change (FC)| > 0.5 were considered as significant DEGs [15]. To construct a clustering hierarchy of DEGs, hierarchical clustering analysis was used. The heat-map package was used for hierarchical cluster analysis. DEGs were clustered using Euclidean distance and then dendrograms were generated. Statistical analyses were performed for each dataset, and the intersecting portions were identified.

### 2.4. Enrichment Analyses of Differentially Expressed Genes

The Database for Annotation Visualization and Integrated Discovery (DAVID) [16] (http://david.abcc.ncifcrf.gov/, accessed on 21 September 2020.) was applied to complete GO (http://geneontology.org/, accessed on 21 September 2020.) [17] terms annotation analyses to identify the biological function of DEGs. Based on KEGG [18] (http://www.kegg.jp/, accessed on 22 September 2020.), pathway enrichment analysis of DEGs was performed using KOBAS [19] (http://kobas.cbi.pku.edu.cn/, accessed on 22 September 2020.). A count ≥ 2 and a *p* < 0.05 were defined as the threshold.

### 2.5. PPI Network and Module Analysis

To predict the interaction between proteins encoded by DEGs with the search tool for the retrieval of interacting genes/proteins [20] (STRING, http://string-db.org/, accessed on 26 September 2020.) database (version 11.0), PPI network was created by Cytoscape software [21] (version 3.8.0) and confidence degree of 0.15 was selected. The resulting network is shown as nodes and edges. Cytoscape software’s plug-in Cytohubba was applied to screen the first 10 nodes by degree filter for visualization [22]. High-degree differential genes were considered as hub genes, which have a decisive role in exercise-mediated regulation.

Proteins in the same module usually perform the same biological function. In this study, the modules were analyzed using the Cytoscape software’s plug-in MCODE [23] with the default parameters “Include Loops: false”, “Degree Cutoff: 2”, “Node Score Cutoff: 0.2”, “Haircut: true”, “Fluff: false,” “K-Core: 2” and “Max. Depth from Seed: 100” in the PPI network. We next mined modules with a score > 15. Thereafter, STRING was used to perform annotation analyses of these genes in these modules, and KEGG pathways were identified by KOBAS.

### 2.6. TF Target Regulation Prediction

The iRegulon [24] function in Cytoscape software can predict critical regulators and construct transcription factor regulatory networks. The plug-in iRegulon in Cytoscape software was used to predicted the TFs of hub genes. The minimum identity between orthologous genes was set to 0.05; the maximum false discovery rate on motif similarity was set to 0.001, and the normalized enrichment score > 4.0 was retained.

## 3. Results

### 3.1. Normalization and DEGs Screening

The Bayesian method of the limma R package was used for the normalization of the GSE51835 gene expression dataset, and the results are shown in Figure 1. The DEGs were analyzed with the limma package, and 146 DEGs (95 upregulated and 51 downregulated DEGs) were identified. Numerous differentially expressed genes from the pre-exercise and after-exercise groups of these datasets is shown in Figure 2. Figure 3 shows the hierarchical clustering heat map of the top 50 DEGs.

### 3.2. Enrichment Analyses of DEGs

The DAVID database was used to perform GO functional annotation analyses of the previously identified DEGs. The top 5 GO terms in each group are shown in Table 1. Results of GO functional annotation analysis included three groups: biological process (BP), cellular composition (CC), and molecular function (MF). In the BP group, DEGs were mainly enriched in cellular response to hydrogen peroxide, response to unfolded protein, negative regulation of cell proliferation, cellular response to laminar fluid shear stress, and positive regulation of protein metabolic process. For cellular composition, DEGs were mostly enriched in nucleus, membrane-bounded organelle, intracellular membrane-bounded organelle, cytoplasm, and nucleoplasm. MF analysis showed that DEGs were primarily enriched in chaperone binding, protein binding, transcription factor activity sequence-specific DNA binding, Hsp70 protein binding, and ubiquitin binding.

The result of KEGG pathway analysis indicated that DEGs were mainly involved in the FoxO signaling pathway, protein processing in endoplasmic reticulum, influenza A, the ErbB signaling pathway, and the MAPK signaling pathway, as shown in Table 2. The results of the KEGG network were visualized using Cytoscape software and are presented in Figure 4.

### 3.3. PPI Network and Module Analysis

The STRING database was applied to structure a PPI network with 146 DEGs (95 upregulated and 51 downregulated DEGs). The result was downloaded and analyzed using Cytoscape software. A PPI network was created after removing uncombined nodes. As shown in Figure 5, the network contains 115 nodes and 741 edges. *TNF*, *DUSP1*, *ATF3*, *CXCR4*, *NR4A1*, *BHLHE40*, *CDKN1B*, *SOCS3*, *TNFAIP3*, and *MCL1* were identified as the top 10 hub genes based on their degree values (see Figure 6 and Table 3).

Furthermore, apply MCODE (version 1.6.1) in PPI network to search cluster modules. We observed that this module contains 21 nodes and 157 interaction pairs, as shown in Figure 7. Then, STRING database and KOBAS were used for GO functional annotation analysis and KEGG pathway analysis on all genes in this module respectively. The result of GO functional annotation analysis indicated that genes were significantly enriched in negative regulation of cellular process (ontology: BP), RNA polymerase II regulatory region sequence-specific DNA binding (ontology: MF) and nucleus (ontology: CC). These results are shown in Table 4. Moreover, the genes were mainly enriched in the IL-17 signaling pathway, the MAPK signaling pathway, the TNF signaling pathway, Osteoclast differentiation, and apoptosis, as is shown in Table 5.

### 3.4. TF Target Regulation Prediction

We then sought to predict and analyze TF target gene relationship pairs using the iRegulon plug-in in Cytoscape software. As shown in Figure 8, 4 transcription factors were identified by a normalized enrichment score > 4: *CREM*, *FOXJ3*, *HSF1*, and *CHD1*.

## 4. Discussion

Monocytes are an essential element of immune systems and play a fundamental role in diseases associated with chronic inflammation, such as atherosclerosis [25]. Recent studies suggest that physical exercise can affect the physiological functions of monocytes [26]. To better understand the effect of exercise on monocytes, we analyzed monocyte samples’ gene expression profile pre- and post-exercise. In this study, 95 upregulated and 51 downregulated DEGs associated with exercise and monocytes were identified. The top potential hub genes (*TNF*, *DUSP1*, *ATF3*, *CXCR4*, *NR4A1*, *BHLHE40*, *CDKN1B*, *SOCS3*, *TNFAIP3*, and *MCL1*) were identified and may play an essential role in the effects of exercise on monocytes. These DEGs were mainly enriched in cellular response to hydrogen peroxide and unfolded protein, negative regulation of cell proliferation, and positive regulation of protein metabolic process. In the KEGG analysis, these DEGs enriched in the FoxO signaling pathway, protein processing in endoplasmic reticulum, influenza A, the ErbB signaling pathway, and the MAPK signaling pathway. Moreover, the most significant module analysis indicated that the MAPK signaling pathway, IL-17 signaling pathway, TNF signaling pathway, osteoclast differentiation, and apoptosis were mainly involved. On the basis of our TF-DEGs regulating network, *CHD1*, *TBPL2*, *BCL6*, *ATF3*, *MEF2C*, and *FOXJ2* were considered to be key regulators.

In this study, we found that TNF may be a key gene and significantly enriched in immunomodulatory functions. TNF is a member of the TNF/TNFR cytokine superfamily, which acquires major importance in the immune system’s maintenance and homeostasis, inflammatory response, and host defense. However, dysregulation of TNF leads to chronic inflammation by activating the nuclear factor-kappa B (NF-κB) signaling pathway, and is associated with several human inflammatory pathologies and diseases, such as cancer, atherosclerosis, and type-2 diabetes. Hence, TNF neutralization suppresses chronic inflammation and attenuates inflammatory pathology. It has been shown that TNF can induce tumor necrosis factor-induced protein 3 (TNFAIP3) expression. The protein encoded by TNFAIP3 gene is critical for limiting inflammation by terminating endotoxin- and TNF-induced NF-κB activation. Our study revealed that expression of TNF was downregulated, whereas TNFAIP3 upregulated its expression after high-intensity exercise, which can inhibit the activation of the NF-κB signaling pathway and enhances the physiological functions of monocytes.

DUSP1 is a member of the dual-specificity phosphatase family [27]. DUSP1 has been recognized as a regulator of kinases, and exerts its effects by dephosphorylation of mitogen-activated protein kinase (MAPK), extracellular-signal-regulated kinase (ERK), and c-Jun N-terminal kinase (JNK) [28]. Research studies have indicated that JNK, MAPK, and ERK signaling molecules are critical in inflammatory signaling pathways [29]. Thus, DUSP1 can change the expression of crucial mediators of the innate immune response and regulates immune homeostasis via regulating the activation of MAPK, ERK, and JNK. In this study, the expression of human monocyte DUSP1 was upregulated after high-intensity exercise, which may indicate that exercise inhibits the MAPK signaling pathway by promoting DUSP1 expression, thereby alleviating the expression of inflammatory mediators and improving the prognosis of inflammatory diseases [30].

CXCR4 is a chemokine receptor that plays a crucial role in the trafficking and homeostasis of immune cells such as T lymphocytes [31]. Furthermore, CXCR4 also promotes the homing and retention of hematopoietic stem and progenitor cells in stem cell niches of the bone marrow [32]. Page et al.’s studies have suggested that CXCR4 impacts cardiovascular development [33]. Based on these findings, we suspected that CXCR4 plays a significant role in chronic inflammatory disease treatment and prevention, such as atherosclerosis. However, a further in-depth study is required to gain better knowledge of the potential role of CXCR4 in exercise.

NR4A1 was observed to be significantly upregulated after exercise. The proteins encoded by NR4A1 belong to an orphan member of the nuclear steroid/thyroid receptor superfamily [34]. Its receptor is expressed abundantly in lymphoid organs and tissues, and regulates B and T lymphocytes [35]. Hanna found that the deletion of NR4A1 in monocytes will promote activation of toll-like receptor signaling, indicating that NR4A1 is a potential target for regulating the immune function of monocytes [36]. In addition, Freire indicated that NR4A1 involved in the regulation of pro-inflammatory cytokines was induced by NF-κB and inhibits NF-κB signal activation [37]. Sabaratnam observed that exercise induces the release of NR4A1, modulating the immune responses in people with type-2 diabetes [38]. Similarly, we also found that NR4A1 was significantly upregulated in individuals after exercise. All together, we speculated that exercise might exert beneficial effects on human health by eliminating excessive pro-inflammatory cytokines.

In our TF-DEGs regulatory network, *FOXJ2* was involved in regulating genes related to immunity, such as *DUSP1*, *TNF*, and *SOCS3*. The protein encoded by FOXJ2 participates in several cellular physiological processes, such as cell cycle, cell proliferation, and apoptosis [39,40]. Additionally, we found that SOCS3 was regulated by FOXJ2 through exercise. Studies have previously shown that SOCS3 plays a vital role as an intracellular negative regulator of inflammation by suppressing STAT3 activation [41]. Furthermore, the expression of SOCS3 is inducible in inflammatory tissue [42]. This study found that GO functional annotation analyses related to SOCS3 were enriched mainly in inflammatory response, protein binding, apoptotic process, and cytoplasm. In studies on arthritis, SOCS3 was involved in defense against inflammation and articular tissue repair [43,44]. Overall, we found that SOCS3 plays an essential role in exercise via regulating the expression of inflammation-related genes in immune cells. 

In the present study, we found that the most significantly enriched pathway was the FoxO signaling pathway. The proteins encoded by FoxO genes are a family of transcription factors [45]. In mammals, there are four members: *FOXO1*, *FOXO3a*, *FOXO4*, and *FOXO6*. The FoxO signaling pathway is involved in the regulation of cell cycle, apoptosis and metabolism [46]. Moreover, FoxO plays an important role in the functions of immune-relevant cells [47,48,49]. The relationship between FoxO and inflammation has been described in inflammatory arthritis [50], systemic lupus erythematosus [51], and atherosclerosis [52]. Our study demonstrated that DEGs mainly participated in the FoxO signaling pathway, showing that inflammation-related DEGs enriched in the FoxO signaling pathway were adjusted by exercise. We assume that exercise may play an active role in determining the balance of inflammation state. 

Studies on gender differences in the effects of high-intensity exercise on monocytes gene expression are strikingly scarce. However, it has been reported that monocyte counts and proportion of non-classical monocytes are different amongst men and women under physiological conditions [53]. These differences may be attributed to the effects of estrogen. An increase in estrogen may be associated with the decreased levels of monocytes [54]. PBMCs co-cultured with estrogen had altered expression of toll-like receptor 3 (TLR3), toll-like receptor 7 (TLR7), toll-like receptor 8 (TLR8), toll-like receptor 9 (TLR9), but not toll-like receptor 2 (TLR2), toll-like receptor 4 (TLR4) and the lipopolysaccharide (LPS) receptor [55]. There is evidence that exercise can improve the female’s estrogen level [56]. Furthermore, exercise decreased monocyte counts in overweight women [57]. We therefore speculate that exercise may modulate monocytes by altering the female’s estrogen level. However, further detailed studies should be done to verify this hypothesis. 

Radom-Aizik et al.’s study showed that high-intensity exercise significantly altered the expression of genes and microRNAs that were likely to regulate monocytes’ participation in anti-inflammatory and anti-atherogenic pathway and thus promote vascular health [12]. In this study, we screened for potential hub genes and candidate signaling pathways associated with the effects of high-intensity exercise on human monocytes. We found that TNF, CXCR4 and SOCS3 are key genes responsible for the exercise-induced anti-inflammatory effects in monocytes. Besides, CXCR4 may participate in the exercise-induced anti-atherogenic effects in monocytes. As for the candidate signaling pathways, we found that the MAPK pathway is highly enriched in the exercise-induced anti-inflammatory effects in monocytes. These results are in line with Radom-Aizik et al.’s study. Notably, we found that the NR4A1, DUSP1, TNFAIP3 and FoxO signaling pathways play vital roles in the exercise-induced immune function of monocytes. Meanwhile, our study indicated that FOXJ2 is a key upstream transcription factor of hub genes.

In this study, we identified several essential genes and signaling pathways involved in anti-inflammation response. Although the gene expression associated with the immune function of monocytes was altered immediately after exercise, this can be a fitness to a reparative response due to the tissue damage induced by exercise and metabolic changes.

However, there are some limitations in the present study. Firstly, the sample size for gene expression profiling analysis was small, which might lead to false positive results. Secondly, we cannot fulfil a follow-up downstream research, which is urgently needed to verify our findings.

## 5. Conclusions

In this study, we identified 95 upregulated and 51 downregulated genes associated with exercise and monocytes. Four essential genes, *TNF*, *DUSP1*, *CXCR4*, and *NR4A1*, and the FoxO signaling pathway, play vital roles in the immune function of monocytes. These findings indicate that high-intensity exercise may enhance resistance to diseases with chronic inflammation by regulating the expression of these genes. However, further experiment of molecular biology studies is needed to confirm the findings of this study.

## Figures and Tables

**Figure 1 healthcare-09-00618-f001:**
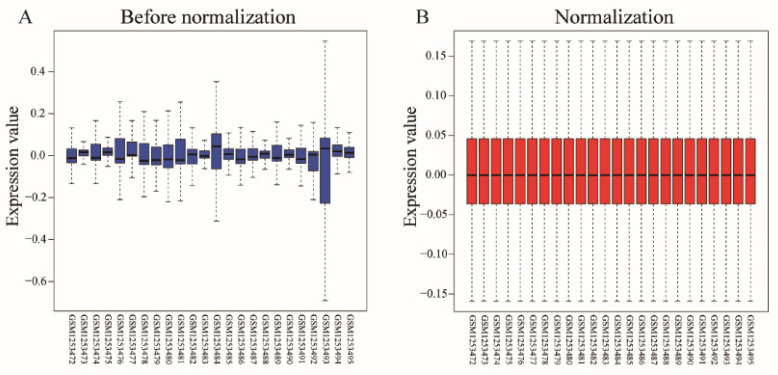
Normalization of gene expression in the GSE51835 dataset.

**Figure 2 healthcare-09-00618-f002:**
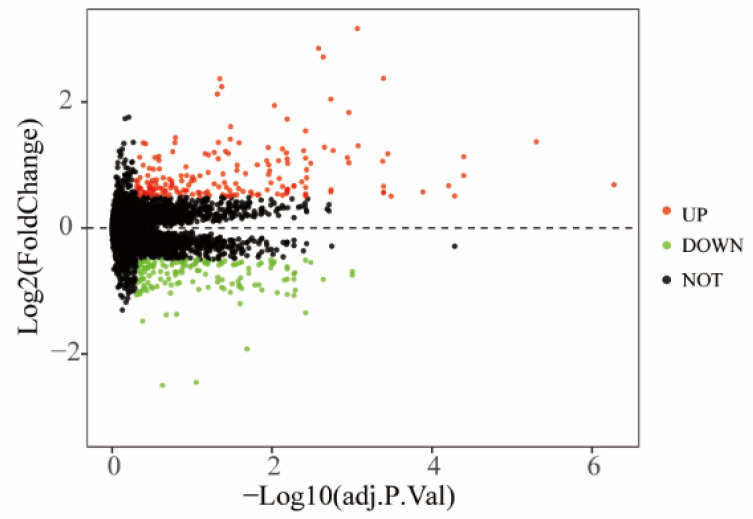
Differentially expressed genes between the two groups of samples. The red points represent upregulated genes. The green points represent downregulated genes. The black spots represent genes with no significant difference.

**Figure 3 healthcare-09-00618-f003:**
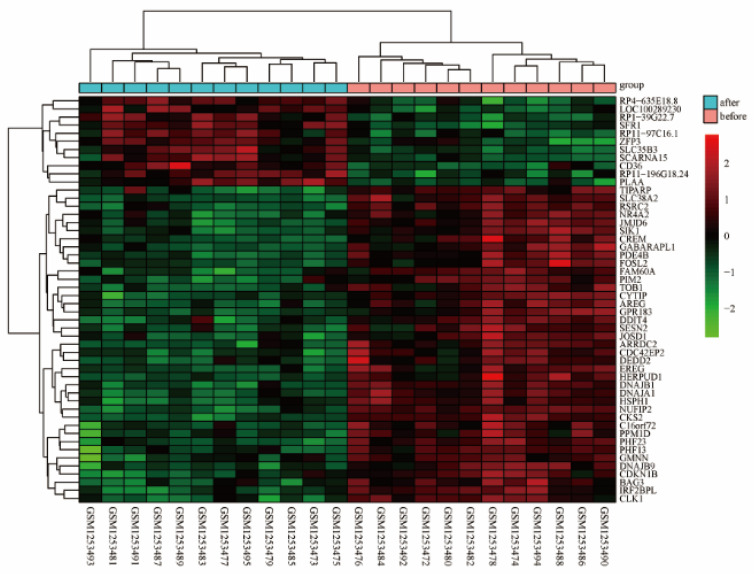
Hierarchical clustering heat map of the top 50 DEGs. Red indicates that gene expression is relatively upregulated, green indicates that gene expression is relatively downregulated, and black indicates no significant changes in gene expression.

**Figure 4 healthcare-09-00618-f004:**
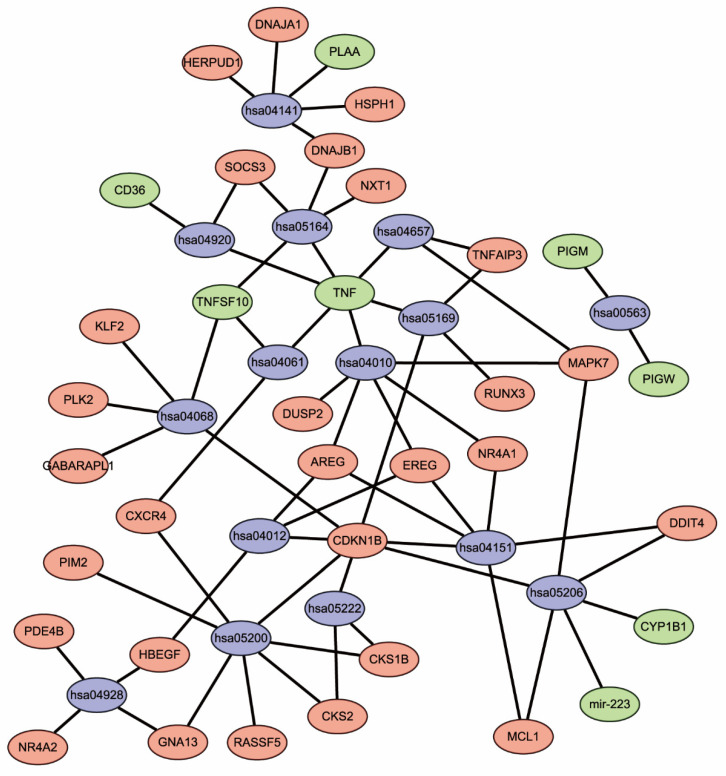
Network diagram of enriched pathways. Blue represents pathways, red represents upregulated DEGs, and green represents downregulated DEGs.

**Figure 5 healthcare-09-00618-f005:**
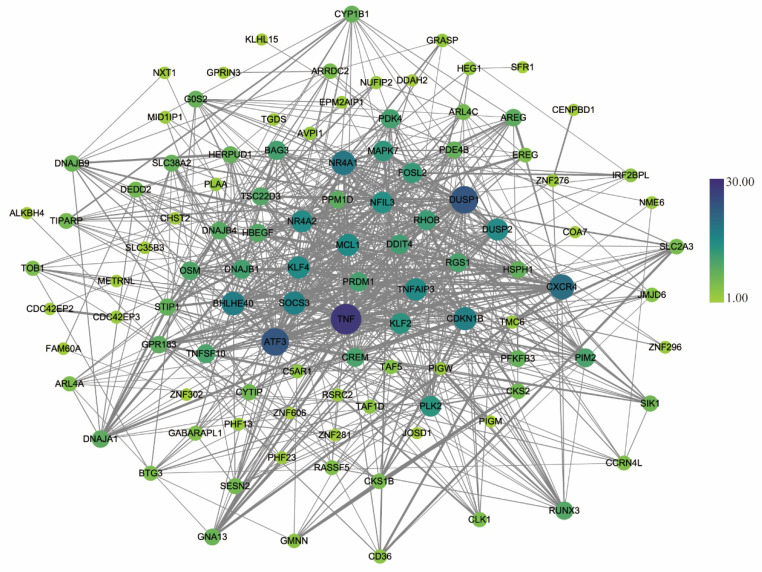
PPI network. Nodes represent genes, and edges represent the interaction of genes. The size of a node is positively correlated with the degree, and the edge between two nodes is positively correlated with the combined score.

**Figure 6 healthcare-09-00618-f006:**
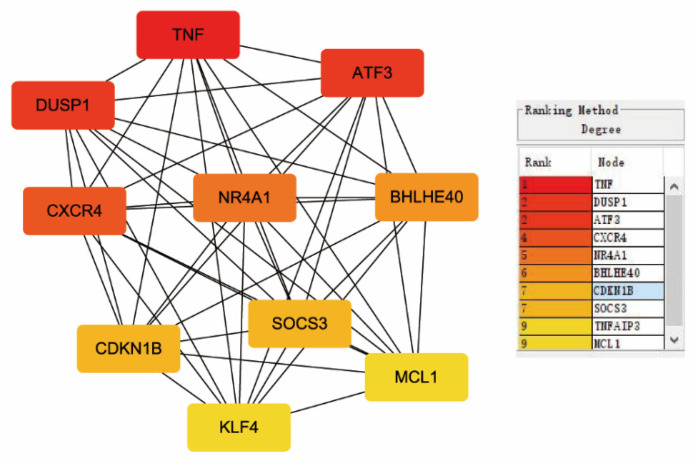
The top 10 hub genes screened by degree values. Different colors indicate the rank of degree.

**Figure 7 healthcare-09-00618-f007:**
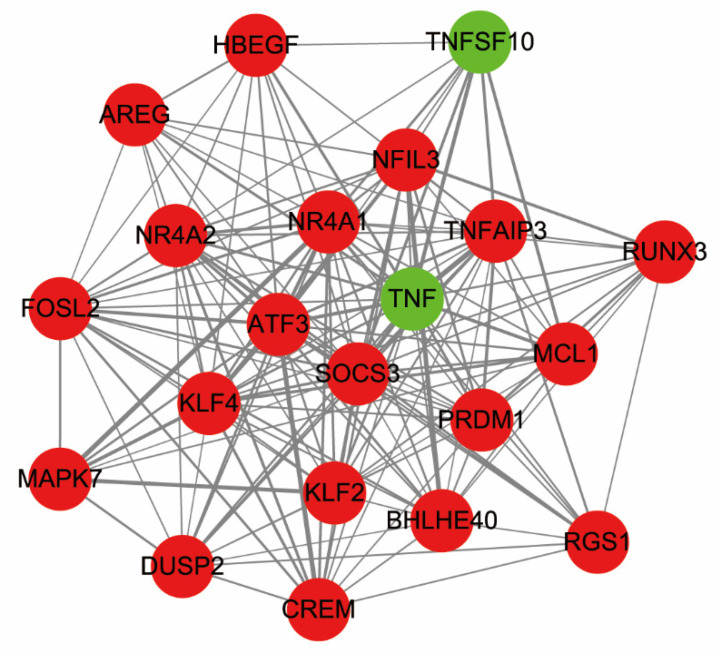
Module diagram of interaction network. Red and green represent genes from upregulation to downregulation.

**Figure 8 healthcare-09-00618-f008:**
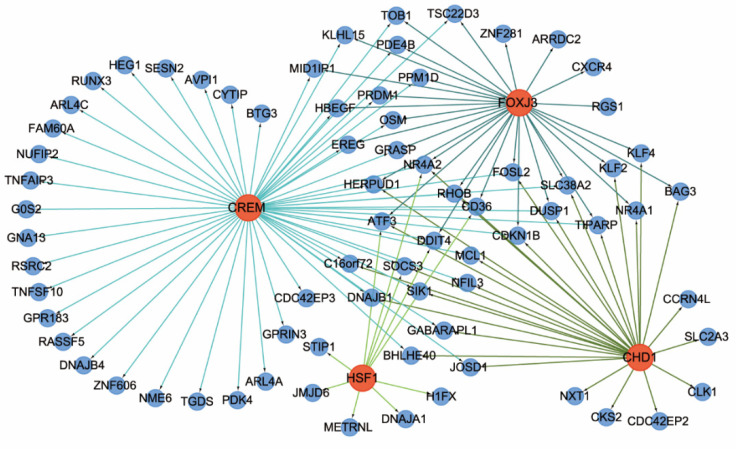
A map of TF target network regulation. Red represents a TF and blue represents the genes; and the arrows connecting the lines indicate the regulatory direction. TF, transcription factor.

**Table 1 healthcare-09-00618-t001:** Top 5 GO enrichment significance items in different functional groups.

GO-ID	Term	Gene Counts	*p*-Value
GO-BP terms			
GO:0070301	cellular response to hydrogen peroxide	6	2.81 × 10^−5^
GO:0006986	response to unfolded protein	5	1.35 × 10^−4^
GO:0008285	negative regulation of cell proliferation	10	8.43 × 10^−4^
GO:0071499	cellular response to laminar fluid shear stress	3	0.001058
GO:0051247	positive regulation of protein metabolic process	3	0.001058
GO-CC terms			
GO:0005634	Nucleus	52	6.68 × 10^−5^
GO:0043227	membrane-bounded organelle	67	0.00014
GO:0043231	intracellular membrane-bounded organelle	62	0.0009
GO:0005737	Cytoplasm	45	0.003735
GO:0005654	Nucleoplasm	26	0.017733
GO-MF terms			
GO:0051087	chaperone binding	6	1.70 × 10^−4^
GO:0005515	protein binding	75	3.58 × 10^−4^
GO:0003700	transcription factor activity sequence-specific DNA binding	16	0.001167
GO:0030544	Hsp70 protein binding	4	0.001208
GO:0043130	ubiquitin binding	5	0.001571

Notes: BP, biological process; CC, cellular component; MF, molecular function.

**Table 2 healthcare-09-00618-t002:** Kyoto Encyclopedia of Genes and Genomes (KEGG) pathway analysis of DEGs.

Pathway	ID	Gene Counts	*p*-Value	Genes
FoxO signaling pathway	hsa04068	5	1.69 × 10^−5^	*PLK2, CDKN1B, TNFSF10, GABARAPL1, KLF2*
Protein processing in endoplasmic reticulum	hsa04141	5	4.75 × 10^−5^	*DNAJB1, PLAA, DNAJA1, HERPUD1, HSPH1*
Influenza A	hsa05164	5	5.02 × 10^−5^	*DNAJB1, TNFSF10, NXT1, SOCS3, TNF*
ErbB signaling pathway	hsa04012	4	5.46 × 10^−5^	*CDKN1B, EREG, HBEGF, AREG*
MAPK signaling pathway	hsa04010	6	7.17 × 10^−5^	*AREG, NR4A1, TNF, MAPK7, DUSP2, EREG*
MicroRNAs in cancer	hsa05206	6	7.71 × 10^−5^	*DDIT4, CDKN1B, MAPK7, CYP1B1, mir-223, MCL1*
Parathyroid hormone synthesis, secretion and action	hsa04928	4	0.000124	*NR4A2, HBEGF, PDE4B, GNA13*
PI3K-Akt signaling pathway	hsa04151	6	0.00019	*AREG, DDIT4, CDKN1B, NR4A1, MCL1, EREG*
Pathways in cancer	hsa05200	7	0.000249	*PIM2, CDKN1B, CXCR4, RASSF5, CKS1B, GNA13, CKS2*
Adipocytokine signaling pathway	hsa04920	3	0.000619	*TNF, SOCS3, CD36*
Epstein–Barr virus infection	hsa05169	4	0.001303	*CDKN1B, TNF, TNFAIP3, RUNX3*
IL-17 signaling pathway	hsa04657	3	0.001425	*TNF, MAPK7, TNFAIP3*
Small-cell lung cancer	hsa05222	3	0.001425	*CDKN1B, CKS1B, CKS2*
Viral protein interaction with cytokine and cytokine receptor	hsa04061	3	0.001744	*TNFSF10, TNF, CXCR4*
Glycosylphosphatidylinositol (GPI)-anchor biosynthesis	hsa00563	2	0.001751	*PIGM, PIGW*

**Table 3 healthcare-09-00618-t003:** Top 10 hub genes with higher degree of connectivity.

Gene Symbol	Gene Description	Degree
TNF	Tumor necrosis factor	56
DUSP1	Dual-specificity protein phosphatase 1	47
ATF3	Cyclic AMP-dependent transcription factor ATF-3	47
CXCR4	C-X-C chemokine receptor type 4	40
NR4A1	Nuclear receptor subfamily 4 group A member 1	37
BHLHE40	Class E basic helix-loop-helix protein 40	35
CDKN1B	Cyclin-dependent kinase inhibitor 1B	34
SOCS3	Suppressor of cytokine signaling 3	34
TNFAIP3	Tumor necrosis factor alpha-induced protein 3	31
MCL1	Induced myeloid leukemia cell differentiation protein Mcl-1	31

**Table 4 healthcare-09-00618-t004:** Top 5 GO enrichment significance items in Module.

GO-ID	Term	Gene Counts	*p*-Value
GO-BP terms			
GO:0048523	negative regulation of cellular process	20	3.59 × 10^−9^
GO:0031324	negative regulation of cellular metabolic process	16	2.26 × 10^−8^
GO:0010604	positive regulation of macromolecule metabolic process	17	2.77 × 10^−8^
GO:0009968	negative regulation of signal transduction	12	7.14 × 10^−8^
GO:0051172	negative regulation of nitrogen compound metabolic process	15	7.14 × 10^−8^
GO-CC terms			
GO:0005634	Nucleus	16	0.0185
GO:0044428	nuclear part	12	0.0337
GO-MF terms			
GO:0000977	RNA polymerase II regulatory region sequence-specific DNA binding	9	7.56 × 10^−7^
GO:0001012	RNA polymerase II regulatory region DNA binding	9	7.56 × 10^−7^
GO:0044212	transcription regulatory region DNA binding	10	7.56 × 10^−7^
GO:0043565	sequence-specific DNA binding	10	9.79 × 10^−7^
GO:0003677	DNA binding	13	2.79 × 10^−6^

**Table 5 healthcare-09-00618-t005:** Kyoto Encyclopedia of Genes and Genomes (KEGG) pathway analysis of Module.

Pathway	ID	Gene Counts	*p*-Value	Genes
MAPK signaling pathway	hsa04010	5	7.84 × 10^−8^	*AREG, NR4A1, MAPK7, TNF, DUSP2*
IL-17 signaling pathway	hsa04657	3	6.23 × 10^−6^	*TNF, MAPK7, TNFAIP3*
TNF signaling pathway	hsa04668	3	1.07 × 10^−5^	*TNF, TNFAIP3, SOCS3*
Osteoclast differentiation	hsa04380	3	1.58 × 10^−5^	*TNF, SOCS3, FOSL2*
Apoptosis	hsa04210	3	1.89 × 10^−5^	*TNFSF10, MCL1, TNF*
Fluid shear stress and atherosclerosis	hsa05418	3	2.01 × 10^−5^	*TNF, KLF2, MAPK7*
Necroptosis	hsa04217	3	3.15 × 10^−5^	*TNFSF10, TNF, TNFAIP3*
Influenza A	hsa05164	3	3.45 × 10^−5^	*TNFSF10, TNF, SOCS3*
Epstein–Barr virus infection	hsa05169	3	5.92 × 10^−5^	*RUNX3, TNF, TNFAIP3*
Type 2 diabetes mellitus	hsa04930	2	0.000148	*TNF, SOCS3*

## Data Availability

The data presented in this study are openly available in FigShare at 10.6084/m9.figshare.14638299.

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
