# Peer review of "Identification of Critical Genes and Signaling Pathways in Human Monocytes Following High-Intensity Exercise"

_healthcare, 2021, doi:10.3390/healthcare9060618_

Round 1
Reviewer 1 Report
The article was changed and resubmitted.
The design and objectives of the study are the same as an original article already published: "Impact of brief exercise on circulating monocyte gene and microRNA expression: Implications for atherosclerotic vascular disease, published in the Journal Brain Behav Immun. 2014 Jul; 39: 121–129)". The authors use the database of this publication to add new analyzes, using software STRING and Cytoscape, among other designs of interactions between protein-protein. Although these re-analyzes bring some new data, the article does not have broader objectives and different from those already presented in the original article. Furthermore, since it is a manuscript that strictly analyzes in silico, it does not fall within the scope of the Healthcare Journal. I suggest that the authors submit to other journals of the mdpi series focused on bioinformatics analysis (example, Data Journal or BiomedInformatics), or focused on the area of sports activities (example, Journal of Functional Morphology and Kinesiology).
Author Response
Point 1: The design and objectives of the study are the same as an original article already published: "Impact of brief exercise on circulating monocyte gene and microRNA expression: Implications for atherosclerotic vascular disease, published in the Journal Brain Behav Immun. 2014 Jul; 39: 121–129)". The authors use the database of this publication to add new analyzes, using software STRING and Cytoscape, among other designs of interactions between protein-protein. Although these re-analyzes bring some new data, the article does not have broader objectives and different from those already presented in the original article. Furthermore, since it is a manuscript that strictly analyzes in silico, it does not fall within the scope of the Healthcare Journal. I suggest that the authors submit to other journals of the mdpi series focused on bioinformatics analysis (example, Data Journal or Biomed Informatics), or focused on the area of sports activities (example, Journal of Functional Morphology and Kinesiology).
Response 1:Thank you for your careful consideration of our manuscript and thanks for your suggestion.
Reviewer 2 Report
This manuscript performs a study that aims to detect potential core genes and candidate signaling pathways associated with the effects of high-intensity exercise in human monocytes through bioinformatic analysis. The authors used an in silico study to estimate the relationship between high intensity exercise and gene expression, the authors found the expression of 95 up-regulated and 51 down-regulated genes.
The authors find four essential genes, TNF, DUSP1, CXCR4, and NR4A1, and the FoxO signaling pathway, that play vital roles in monocyte immune function. These data are interesting although the limitations presented by the authors themselves for their interpretation must be taken into account.
Within the discussion the authors present some evidence on different factors that could modulate the effect of exercise on gene expression in monocytes, in this sense I suggest expanding the information of the subjects included in the original study in the methodology section where they are described the subjects.
Author Response
Point 1: Within the discussion the authors present some evidence on different factors that could modulate the effect of exercise on gene expression in monocytes, in this sense I suggest expanding the information of the subjects included in the original study in the methodology section where they are described the subjects.
Response 1: Thank for your constructive suggestion. We have expanded the information of the subjects who participated in the study. The detailed description are as follows:
1.“mean age 26 ± 0.6 years and body mass index 26 ± 0.8kg/m2” (Line 48-49, Page2)
2.“The participants who actively joined in the elite or competitive level of sports, or were suffering from any chronic illness or medication were excluded from the study.” (Line 50-51, Page2)
3. “Subjects were asked to refrain from intense physical activity for 48 h before the exercise challenge.” (Line 4-5, Page 3)
Reviewer 3 Report
The study titled: “Identification of Critical Genes and Signaling Pathways in Hu-man Monocytes Following High-Intensity Exercise” show interesting result related to monocyte gene expression in exercise in males.
As suggestions, in introduction section, the authors need to improve information related to monocytes, as these are myeloid cells in circulation and not all them are differentiated to macrophages, these cells are part of the first line of defence along with neutrophils (which are not mentioned), and participates in pro and anti-inflammatory responses, depend on stimulus. In addition, some references could be added, as “Scheffer DDL, Latini A. Exercise-induced immune system response: Anti-inflammatory status on peripheral and central organs. Biochim Biophys Acta Mol Basis Dis. 2020 Oct 1;1866(10):165823.”, and more important the reference “Radom-Aizik, S.; et al, Impact of brief exercise on circulating monocyte gene and microRNA expression: implications for atherosclerotic vascular disease. Brain, behavior, and immunity 2014, 39, 121-129” should be mention in introduction, due the origin of samples and the results related to MAPK, SOCS3 and TNF.
In materials and methods is important to mention which class of monocytes are studied (CD14+CD16- or CD14+CD16+ or both)
In discussion section the authors could mention that the results represent gene expression of monocytes post immediately exercise, which not necessary reflects a stable state, as well that this can be a fitness to a reparative response due tissue damage in exercise and metabolic changes, more than an anti-inflammatory response, that could be possible in a long term period of constant exercise.
Author Response
Point 1: As suggestions, in introduction section, the authors need to improve information related to monocytes, as these are myeloid cells in circulation and not all them are differentiated to macrophages, these cells are part of the first line of defence along with neutrophils (which are not mentioned), and participates in pro and anti-inflammatory responses, depend on stimulus. In addition, some references could be added, as “Scheffer DDL, Latini A. Exercise-induced immune system response: Anti-inflammatory status on peripheral and central organs. Biochim Biophys Acta Mol Basis Dis. 2020 Oct 1;1866(10):165823.”, and more important the reference “Radom-Aizik, S.; et al, Impact of brief exercise on circulating monocyte gene and microRNA expression: implications for atherosclerotic vascular disease. Brain, behavior, and immunity 2014, 39, 121-129” should be mention in introduction, due the origin of samples and the results related to MAPK, SOCS3 and TNF.
Response 1: Thank you for your insightful opinions. We have added information related to monocytes in the introduction, as shown below: Monocytes, which originate from progenitors in the bone marrow, are part of the first line of immune defense along with neutrophils and circulate in the vasculature, bone marrow, and spleen. In the normal course of development, monocytes migrate into peripheral tis-sues and differentiate into dendritic cells or macrophages depending on the local cytokine environment. During pathogen challenge, monocytes are mobilized from the bone mar-row and recruited to sites of inflammation, where they carry out their respective functions in promoting inflammation or anti-inflammatory responses depending on stimulus style. (Line 43-44, Page 2; Line 1-6, Page 3)
We have added the references into the introduction as you suggested. (Line 10-11, 27,35, Page 2);
Point 2: In materials and methods is important to mention which class of monocytes are studied (CD14+CD16- or CD14+CD16+ or both).
Response 2: Thank you for your insightful opinions. The classes of monocytes which are studied have been added into the materials and methods section as you suggested, as shown below: Monocytes were isolated from blood, including classical (CD14++/CD16−), nonclassical (CD16++/CD14+) and intermediate (CD16+/CD14++) monocytes. (Line 2-4, Page 3)
Point 3: In discussion section the authors could mention that the results represent gene expression of monocytes post immediately exercise, which not necessary reflects a stable state, as well that this can be a fitness to a reparative response due tissue damage in exercise and metabolic changes, more than an anti-inflammatory response, that could be possible in a long-term period of constant exercise.
Response 3: Thank you for your constructive suggestion. We have added this idea into the manuscript as follows: In this study, we identified several essential genes and signaling pathways involved in anti-inflammation response. Although the gene expression associated with the immune function of monocytes was altered immediately after exercise, this can be a fitness to a reparative response due to the tissue damage induced by exercise and metabolic changes. (Line 1-4, Page 13)
Round 2
Reviewer 1 Report
I suggest that the authors submit to other journals of the mdpi series focused on bioinformatics analysis (example, Data Journal or BiomedInformatics), or focused on the area of sports activities (example, Journal of Functional Morphology and Kinesiology).
Author Response
Point 1: I suggest that the authors submit to other journals of the mdpi series focused on bioinformatics analysis (example, Data Journal or BiomedInformatics), or focused on the area of sports activities (example, Journal of Functional Morphology and Kinesiology).
Response 1: Thank you for your careful consideration of our manuscript and thanks for your suggestion.
Research fields of Healthcare journal include but are not limited to:
1.Chronic care (Health assessment; Laboratory and diagnostic procedures; Medication management; Disease prevention; Early diagnosis; Treatment and comprehensive strategies; Morbidity and mortality; Long term outcomes; Scoring systems and outcome prediction; Socio-economic burden of chronic care)
2.Critical care (Emergency, perioperative and intensive care; Pediatric/neonates care; Medical imaging, monitoring, support)
3.Advanced inpatients care (Advanced medical investigation and treatment; Experimental medicine; Uncommon diagnostic or surgical procedures)
4.Health informatics (electronic and online based clinical studies and medical research)
5.Health care materials
6.Mental health
7.Nursing (Treatment for a short period of time for a brief but serious illness, injury or other health condition)
Our research falls within “4. Health informatics (electronic and online based clinical studies and medical research)”. So we submitted our manuscript to Healthcare, and hope that it is suitable for publication in the journal.
This manuscript is a resubmission of an earlier submission. The following is a list of the peer review reports and author responses from that submission.
Round 1
Reviewer 1 Report
Comments
General: In the research, the authors present evidence of the identification of central genes, potentials associated with the effects of high intensity exercise in monocytes through bioinformatic analysis and to be able to elucidate the underlying molecular mechanisms.
The authors present good information about a study that included the dataset of 12 monocyte samples from two groups of pre-exercise and post-exercise individuals.
Manuscript title: I consider that it is correct.
Abstract: It`s correct, although I suggest to be more clear in presenting the objective of the study.
Introduction: The literature used is pertinent.
Methods: the study was clearly stated. However, there are some concerns such as:
- I consider it necessary for the authors to provide information regarding the provision of the participants' informed consent and whether this study was approved by a local ethics committee.
- it would be important to provide a characterization of the subjects who participated in the study.
Results: the choice of tables and figures it's correct, I suggest including a table with the characteristics of the subjects
Discussion and Conclusions: I suggest that the authors identify more clearly the strengths and limitations of the study.
References: There were 51 and all appropriate.
Reviewer 2 Report
Review
Title: Identification of Critical Genes and Signaling Pathways in Human Monocytes Following High Intensity Exercise
Monocytes are immune cells necessary for the development of an innate immune response, in addition to activating the adaptive immune system. Monocytes play a fundamental role in the development of chronic inflammation, but it promotes inflammatory resistance in athletes. The present study identified a hub of genes expressed by monocytes and which are associated with the effects of high physical intensity. For this, the authors used a gene expression database (GSE51835). A total of 146 differentially expressed genes were identified (95 over-regulated and 51 under-regulated). Essentially 4 genes (TNF, DUSP1, CXCR4, and NR4A1) and the FoxO signaling pathway play a vital role in monocyte immune function.
This article provides a reanalysis of the data from the original article: Brain Behav Immun. 2014 Jul; 39: 121–129. The methodologies are close, with the exception of the bioinformatics tools that were updated from 2014 to 2020. The results are similar, but the authors forget to mention such data.
If the authors propose a reanalysis of the data from the original article, this should be made clear, and Healthcare Journal does not have an ideal scope for this publication. An editorial letter or a short communication could be submitted to the Journal Brain Behav Immun, where the original article is published; or the choice of one of the mdpi series focused on bioinformatics analysis, as an example: Data Journal.
On the other hand, the study has positive points. Updating bioinformatics analysis using R, building the interaction between genes and proteins using STRING and Cytoscape software. These updates are necessary and of interest to the scientific community that searches for gene expression databases.
Below, some parts of the text will be highlighted, which requires some care because it contains transcripts very close to the original data.
Material and Methods
The study is an analysis in silico, the laboratory part must be removed, or better explained.
Original description text: “GSE51835 - Twelve healthy men (22-30 yr old) performed 10 2-min bouts of cycle ergometer exercise interspersed with 1-min rest at a constant work equivalent to about 82% of VO2max. A baseline blood sample was taken before the and immediately after the exercise. Monocytes were isolated from PBMC using a negative magnetic cell separation methods (Miltenyi Biotec Monocyte Isolation Kit II human #130-091-153 and the autoMACS® Pro Separator). Total RNA was extracted using TRIzol®. For this study we used Affymetrix plus 2 (total of 24 chips).”
Authors text: “2. Materials and Methods 2.1. Exercise intervention from the original experiment Twelve healthy men in the 22–30 age range performed ten 2-min episodes cycle ergometer exercise of 82% VO2max mixed with 1 min rest. Blood samples were collected at the before exercise baseline and immediately after exercise. Monocyte isolation Kit II (Miltenyi Biotec, Bergisch Gladbach, Germany) was used to isolate mononuclear cells from peripheral blood and monocyte subtypes were identified with autoMACS® Pro Separator (Miltenyi Biotec) Flow cytometry. Subsequently, total RNA was extracted using TRIzol® (Gibco BRL Life Technologies, Rockville, MD, USA) reagent.”
Original description text: “The Database for Annotation Visualization and Integrated Discovery (DAVID)[15] (http://david.abcc.ncifcrf.gov/) was applied to complete GO (http://geneontology.org/)[16] terms annotation analyses to identify the biological function of DEGs. Based on KEGG[17] (http://www.kegg.jp/), Pathway enrichment analysis of DEGs was performed using KOBAS[18] (http://kobas.cbi.pku.edu.cn/). A count >= 2 and a P < 0.05 were defined as the threshold.”
Authors text: “The final list of significantly changed probe sets was then additionally analyzed using the functional annotation tools provided by DAVID, the Database for Annotation, Visualization and Integrated Discovery to classify the genes into pathways using the Kyoto Encyclopedia of Genes and Genomes (KEGG) database. Only pathways with Expression Analysis Systematic Explorer (EASE) score ≤ 0.038 are presented in this analysis. EASE score is a modified Fisher exact P value in the DAVID system used for gene-enrichment analysis. EASE score P value = 0 represents perfect enrichment. P value ≤ 0.05 is considered as gene enrichment in a specific annotation category (http://david.abcc.ncifcrf.gov/helps/functional_annotation.html#summary).”
Discussion
Authors need to make clear the similarity of the findings with the original article.
For example, of the 10 genes found by the authors: “TNF, DUSP1, ATF3, CXCR4, NR4A1, BHLHE40, CDKN1B, SOCS3, TNFAIP3, and MCL1 were top 10 potential hub genes”, five of which have already been described by the original article (TNF, DUSP1, CXCR4, NR4A1 and SOCS3).
Reviewer 3 Report
This research article entitled “Identification of Critical Genes and Signaling Pathways in Human Monocytes Following High Intensity Exercise” presents interesting finding on the effects of exercise on monocytes and associated gene expression changes. The article is simple and well written (except some grammatical mistakes and typos). The analysis reported here will be of immense interest to the readers of this journal. However, I have following concerns that if addressed will be helpful:
- This paper evaluates differential gene expression from a publicly available database without any follow up downstream research or verification of the reported findings. As such novelty and reliability of the findings become questionable.
- The only thing to peer review in this article is the codes and methods used for the whole analysis. I know the R packages used here are publicly available. Are the codes used in this study available in general or for review?
- The gene expression profile is from male and female-specific information is missing. This should be discussed in the context of existing literature on male/female differences in immune functions.
- Figure1: Individually labeled graphs should be described in the legend.
- Tables: What each column represents should be described as legends for each table. For example, what does “count” mean in the respective tables? This will bring more clarity for readers not familiar with RNAseq studies.